# Development of Bisphosphonate-Conjugated Antibiotics to Overcome Pharmacodynamic Limitations of Local Therapy: Initial Results with Carbamate Linked Sitafloxacin and Tedizolid

**DOI:** 10.3390/antibiotics10060732

**Published:** 2021-06-17

**Authors:** Emmanuela Adjei-Sowah, Yue Peng, Jason Weeks, Jennifer H. Jonason, Karen L. de Mesy Bentley, Elysia Masters, Yugo Morita, Gowrishankar Muthukrishnan, Philip Cherian, X. Eric Hu, Charles E. McKenna, Frank H. Ebetino, Shuting Sun, Edward M. Schwarz, Chao Xie

**Affiliations:** 1Center for Musculoskeletal Research, University of Rochester Medical Center, Rochester, NY 14642, USA; eadjeiso@UR.Rochester.edu (E.A.-S.); Yue_Peng@URMC.Rochester.edu (Y.P.); jweeks3@u.rochester.edu (J.W.); Jennifer_Jonason@urmc.rochster.edu (J.H.J.); Karen_Bentley@urmc.rochester.edu (K.L.d.M.B.); emaster5@ur.rochester.edu (E.M.); Yugo_Morita@urmc.rochster.edu (Y.M.); Gowri_Shankar@urmc.rochester.edu (G.M.); Edward_Schwarz@urmc.rochster.edu (E.M.S.); 2Department of Biomedical Engineering, University of Rochester, Rochester, NY 14642, USA; 3Department of Pathology, University of Rochester Medical Center, Rochester, NY 14642, USA; 4Department of Orthopaedics, University of Rochester Medical Center, Rochester, NY 14642, USA; 5BioVinc, LLC, Pasadena, CA 91107, USA; ptcherian@biovinc.com (P.C.); eric.hu@biovinc.com (X.E.H.); halebetino@biovinc.com (F.H.E.); 6Department of Chemistry, University of Southern California, Los Angeles, CA 90089, USA; mckenna@usc.edu; 7Department of Chemistry, University of Rochester, Rochester, NY 14642, USA

**Keywords:** osteomyelitis, *Staphylococcus aureus*, antibiotic, bisphosphonate, scanning electron microscopy

## Abstract

The use of local antibiotics to treat bone infections has been questioned due to a lack of clinical efficacy and emerging information about *Staphylococcus aureus* colonization of the osteocyte-lacuno canalicular network (OLCN). Here we propose bisphosphonate-conjugated antibiotics (BCA) using a “target and release” approach to deliver antibiotics to bone infection sites. A fluorescent bisphosphonate probe was used to demonstrate bone surface labeling adjacent to bacteria in a *S. aureus* infected mouse tibiae model. Bisphosphonate and hydroxybisphosphonate conjugates of sitafloxacin and tedizolid (BCA) were synthesized using hydroxyphenyl and aminophenyl carbamate linkers, respectively. The conjugates were adequately stable in serum. Their cytolytic activity versus parent drug on MSSA and MRSA static biofilms grown on hydroxyapatite discs was established by scanning electron microscopy. Sitafloxacin *O*-phenyl carbamate BCA was effective in eradicating static biofilm: no colony formation units (CFU) were recovered following treatment with 800 mg/L of either the bisphosphonate or α-hydroxybisphosphonate conjugated drug (*p* < 0.001). In contrast, the less labile tedizolid *N*-phenyl carbamate linked BCA had limited efficacy against MSSA, and MRSA. CFU were recovered from all tedizolid BCA treatments. These results demonstrate the feasibility of BCA eradication of *S. aureus* biofilm on OLCN bone surfaces and support in vivo drug development of a sitafloxacin BCA.

## 1. Introduction

Bone infection, primarily caused by *Staphylococcus aureus,* remains a scourge in orthopaedic surgery. Although the incidence of infection following primary total joint replacement (TJR) is low (~1%) [1,2,3], reinfection rates are very high (15–40%) after treatment of an infection related to a joint replacement [4,5,6,7,8], which has led to the established paradigm that *S. aureus* infection of bone is incurable [9]. The original standard of care treatment algorithm for implant-associated osteomyelitis was established in the 1970s and involves: (1) removal of the infected implant, (2) extensive surgical debridement of adjacent bone and soft tissues, and (3) filling of the bone void with antibiotic-loaded bone cement (ALBC) [10]. Initial studies on the effectiveness of this algorithm in the 1980s revealed that *S. aureus* was the most commonly encountered organism, and that the five-year success (survival) rate was only 77% [11]. Remarkably, the results from the 2018 International Consensus Meeting (ICM) on Musculoskeletal Infections (MSKI) reported no changes in prosthetic joint infection (PJI) rates, the primary pathogen, treatment algorithm, and poor outcomes since this original standard of care was established half a century ago [1,2,3]. Thus, it is clear that both conceptual innovations to improve our understanding of the problem, and technical innovations to enhance diagnosis and treatment, are needed to improve clinical outcomes from *S. aureus* osteomyelitis.

Currently, the management of PJI is not universally standardized, but generally includes both systemic antibiotics and surgical debridement with exchange arthroplasty in one or two stages. There is great variation in management of the infection using poly(methyl methacrylate) (PMMA) ALBC and short (~weeks) and long (>3 months) courses of parenteral and oral antibiotic therapy, respectively [12]. The choice of antibiotic is usually based on etiologic diagnosis (culture and sensitivity of the infecting organism). ALBC is usually used in two-stage exchange arthroplasty in the form of spacers, beads, rods, and other custom spacers. Commonly used antibiotics include gentamicin, tobramycin, and vancomycin powder depending on the microbial organism [13]. While common, this practice has limitations, including microscopic imperfections in hand mixed and molded spacers that could lead to mechanical failure and/or poor release kinetics [14], which necessitates high loading of antibiotics that could theoretically lead to antibiotic resistance and acute renal failure [13]. Additionally, ALBCs are temporary spacers, and must be removed after the management of infection to allow for bone reconstruction. A recent comprehensive review of ALBC use for prophylaxis in primary TJR indicated that this approach is not supported by scientific evidence or health care savings considerations [15]. Thus, a systemic therapy that specifically targets the bone surface most susceptible to infection following orthopaedic implant surgery (prophylaxis), and the interface between biofilm bacteria and chronically infected bone (treatment), would appear to be an ideal goal for novel antibiotic therapies.

The recent discovery of *S. aureus* colonization of the osteocyte lacuno canalicular network (OLCN) of live bone via transmission electron microscopy (TEM) analyses of infected bone from mice, and bone biopsies from patients with chronic osteomyelitis, has provided new insights into the problems of antibiotic therapy to treat bone infections [16,17,18,19]. Specifically, *S. aureus* within the OLCN is a permanent reservoir of bacteria that cannot be eradicated by host responses or any known treatments short of amputation [9,20,21,22]. While in vivo bromodeoxyuridine labeling studies in mice have demonstrated that orally administered small molecules have access to *S. aureus* at the leading edge of the colony within the OLCN [17], we have also shown that both methicillin-sensitive *S. aureus* (MSSA) [23] and methicillin-resistant *S. aureus* (MRSA) [24] OLCN invasion cannot be prevented or eradicated by combined high-dose local and systemic antibiotics, likely due to its well-known adaptive responses that are associated with persister cells and small colony variants (SCVs) [10]. Thus, OLCN invasion might limit the availability of conventional antibiotics to levels below the minimum inhibitory concentration (MIC) and minimum biofilm eradication concentration (MBEC), and therefore activates biological mechanisms of antibiotic tolerance, such as the doubling of the bacterial cell wall thickness that we observed [23]. Therefore, a bone targeted antibiotic whose local concentration can increase over the MIC and MBEC is highly desirable to treat these deep bone infections.

Due to the overall challenges associated with osteomyelitis treatment, we and others have introduced bone-targeting conjugates to achieve higher or more sustained local therapeutic concentrations of antibiotic in bone while minimizing systemic exposure [25]. Specifically, we proposed conjugation of fluoroquinolone antibiotics to osteoadsorptive bisphosphonates (BPs) as a promising approach due to the long clinical track record of safety of the individual drug constituents, and their advantageous biochemical properties [26,27]. As an initial proof of concept, we evaluated BP-conjugated ciprofloxacin, utilizing a “target and release” chemical strategy in vitro, and in a rat model of implant-associated osteomyelitis [28]. In vitro studies demonstrated strong drug binding to hydroxyapatite, and an effective bactericidal profile with sustained release of the parent antibiotic over time. The in vivo results showed that a single dose of 10 mg/kg (15.6 mumol/kg) conjugate reduced the colony forming units (CFU) in tissues by 99% and was one order of magnitude more potent than the parent antibiotic ciprofloxacin (30 mg/kg, 90.6 mumol/kg) given in multiple doses.

Another poorly recognized limitation of conventional antibiotic administration is the effect of local wound drainage on the pharmacodynamics of the therapy. In addressing the question of why local antibiotic therapies are ineffective, a recent international consensus meeting on ALBC highlighted the fact that there is interstitial fluid flow that drains from the bone infection to the surrounding soft tissues [10], as illustrated in Figure 1. This creates convection in the opposite direction, which could thwart all local antibiotic therapies. Here again, BP-conjugated antibiotics are expected to have an advantage via their accumulation at the bone-bacteria interface. To test this, we conducted a pilot study in which mice with established *S. aureus* osteomyelitis were treated with a BP-conjugated fluorescent probe (AF647-ZOL) [29] and undemineralized histology was performed to assess labeling at the infection site. Due to the limited efficacy of the first generation fluoroquinolone-conjugated BPs on established biofilms [30,31,32], we selected the newer generation fluoroquinolone, such as sitafloxacin, and an oxazolidinone antibiotic, tedizolid, and investigated novel BP and α-hydroxybisphosphonate (HBP) conjugates of those antibiotics, which were synthesized with hydroxyphenylethane carbamate and amino phenylethane carbamate linkers respectively that we believe offer sufficient serum stability to permit delivery of the drugs to the bone surface. Here we present the results of these studies, which provide the first evidence that BP-conjugated drugs given to infected mice concentrate on the cortical surface between the bone and bacteria. We also present evidence of the potent cytolytic activity of sitafloxacin conjugated BP and HBP on MSSA and MRSA static biofilms grown on hydroxyapatite discs in vitro, versus parent antibiotic and tedizolid conjugated BP and HBP.

## 2. Material and Methods

### 2.1. S. aureus Strains and In Vitro Culture

Two different *S. aureus* strains were used: (1) UAMS-1, a widely used MSSA strain that was isolated from an osteomyelitis patient [33], and (2) USA300LAC [34], which is the most prevalent community-acquired MRSA strain. All strains were cultured in tryptic soy broth (TSB) media. For static biofilm formation, the bacteria were grown in 24-well plates as previously described [28].

### 2.2. Animal Surgeries and AF647-ZOL Treatment

Alexa Fluor 647 conjugated to zoledronic acid (AF647-ZOL) is a red fluorescent-BP conjugated probe that has been used to demonstrate BP-conjugated drug bone-targeting potential in animal models [29]. AF647-ZOL was obtained from BioVinc, LLC as a lyophilate and used as received. All in vivo experiments with mice were performed following protocols approved by the University of Rochester Committee on Animal Resources (UCAR 2019-015). The surgical approach was performed as previously described [35], in which a flat stainless steel wire (cross-section 0.2 mm × 0.5 mm; MicroDyne Technologies, Plainville, CT, USA) was contaminated with 10^5^ CFU of MSSA or MRSA from an overnight culture, and surgically implanted through the tibia of six-week-old Balb/c female mice (*n* = 3) (Jackson Labs, Bar Harbor, ME, USA). One week later, the mice received an intraperitoneal injection of AF647-ZOL (0.385 mg/kg) as previously described [29], and the mice were euthanized one week later (day 14 post-op) to assess the infected tibiae via histology.

### 2.3. Histology and Fluorescent Microscopy

Infected tibiae were processed for undemineralized frozen histology. The hindlimbs were cleaned of soft tissue and fixed with gentle agitation in 10% Neutral Buffered Formalin (NBF) for 1.5 h at 4 °C followed by an additional hour at room temperature. Samples were then placed in a 30% sucrose/phosphate buffered saline (PBS) solution overnight at 4 °C prior to embedding into Cryomatrix Frozen Embedding Medium (Thermo Fisher, Houston, TX, USA). Samples were sectioned at 10 mm using the previously described tape transfer method and Cryofilm Type 2C [36]. Tape sections were adhered to glass slides, tissue side up, using a solution of 1% chitosan/0.25% acetic acid and allowed to cure for 48 h at 4 °C prior to staining.

Brown–Brenn modified Gram stain to visualize Gram positive bacteria on serial sections was performed as previously described [24]. Immunofluorescent staining was performed as previously described [24], but without antigen retrieval methods. The tissue was blocked in at room temperature in 5% normal goat serum (NGS) in 0.3% Triton-X100 TBS for 40 mins, and then slides were incubated with polyclonal antibody for *S. aureus* (1:100, Invitrogen, Cat#: PA1-7246) in 5% NGS in 0.3% Triton-X100 TBS at 4 °C overnight. Next, anti-rabbit FITC conjugated secondary antibody (1:400, Invitrogen, Cat#: A-11008) in 5% NGS in 0.3% Triton-X100 TBS was added to sections for 1 hr at room temperature. Finally, sections were counterstained with nuclear stain DAPI and mounted with ProLong Gold Antifade Mountant (Life Technologies, Eugene, OR, USA). Stain specificity was validated by incubating with the secondary antibody only (data not shown). Slides were either imaged using a VS120 Virtual Slide Microscope (Olympus, Waltham, MA, USA) for abscess imaging or via confocal laser scanning microscopy (CLSM) for sequestra imaging. CLSM was performed using an inverted Olympus FV 1000 microscope using a 60× oil immersion objective with 0.5 µm slices. Z-stack images were processed using ImageJ to create max-intensity z-projections.

### 2.4. Hydroxyapatite (HA) Discs

Test articles HA discs (3D Biotek, LLC. Bridgewater, NJ, USA) received directly from the manufacturer with sterile package. It is a bone mineral-like material disc compatible for 96-well culture dish with diameter ~5 mm, and thickness ~1.6 mm. The quality of the manufactured HA discs was checked by means of scanning electron microscopy (SEM) as illustrated in Figure 2A–C.

### 2.5. Synthesis and Characterization of BP Conjugated Antibiotics (BCAs)

The chemical structures of sitafloxacin, tedizolid, and the various BP and HBP conjugates used in this study are described in Figure 3 (Bisphosphonate-carbamate-sitafloxacin: BCS; α-hydroxybisphosphonate-carbamate-sitafloxacin: HBCS; Bisphosphonate-carbamate-tedizolid: BCT; α-hydroxybisphosphonate-carbamate-tedizolid: HBCT; Bisphosphonate-ester-tedizolid (BET)). The BPs used for conjugation are: (2-(4-hydroxyphenyl)ethane-1,1-diyl)bis(phosphonic acid) (HPBP); (2-(4-aminophenyl)ethane-1,1-diyl)bis(phosphonic acid) (APBP); 4-(2,2-diphosphonoethyl)benzoic acid (CPBP); and the HBPs used for conjugations are: (1-hydroxy-2-(4-hydroxyphenyl)ethane-1,1-diyl)bis(phosphonic acid) (HPHBP); (2-(4-aminophenyl)-1-hydroxyethane-1,1-diyl)bis(phosphonic acid) (APHBP). All of these compounds were synthesized similarly according to procedures as originally reported in Sedghizadeh et al. [28], and Sun et al. [25]. The compounds were characterized by NMR, mass spectrometry, combustion elemental analysis and HPLC, demonstrating >95% purity.

### 2.6. In Vitro MSSA and MRSA Biofilm Eradication Assays

The dose-dependent efficacy of sitafloxacin and tedizolid conjugated BP and HBP versus their parent antibiotic was assessed using an in vitro HA disc assay described in Figure 4. Briefly, 1 mL of tryptic soy broth (TSB) containing 10^7^ Colony-forming units (CFU) of a bioluminescent MSSA strain (Xen36(Luc)) or MRSA strain (USA300LAC::Luc) was added to each well of a 24-well plate, and HA discs were placed in each well and incubated for 24 h at 37 °C. After 24 h, HA discs were washed three times in 1 mL of sterile PBS for 5 min in a gentle rocker shaker. Then, HA discs were incubated in 800, 400, 200, 100, 50, 25, 10, 5, or 1 mg/L of HBCS, BCS, Sitafloxacin Hydrate, HPHBP, or HPBP, for 24 h at 37 °C. The discs were then rinsed to remove non-bound bacteria, and subjected to vigorous vortex and sonication to collect adherent bacteria from the HA discs. CFU were determined using the Miles and Misra method, in which serial dilutions were cultured on modified TSB agar plates, and quantified after 24 h incubation at 37 °C. This protocol was repeated using HBCT, BCT, Tedizolid, and BET with concentration of 100 and 800 mg/L.

### 2.7. Scanning Electron Microscopy (SEM)

SEM was performed as previously described [35]. Briefly, the HA disks from the in vitro studies were placed into 24 well plates, fixed in 2.5% glutaraldehyde/4% paraformaldehyde in 0.1 M cacodylate overnight and post-fixed in buffered 1% osmium tetroxide. A pipet tip was placed against the wall of the wells, for fluid exchange or removal to reduce disruption of biofilm during dehydration in a graded series of ethanol to 100%. The disks were then critically point dried, mounted onto aluminum stubs and sputter coated with gold prior to imaging using a Zeiss Auriga FE SEM. Three SEM micrographs per sample group were randomly chosen for descriptive analysis.

## 3. Results

### 3.1. Systemically Delivered AF647-ZOL Accumulates at the Bone-Bacteria Interface

To assess the feasibility of BP and HBP-conjugated antibiotic “target and release” therapy for chronic osteomyelitis, we performed a pilot study to assess the in vivo labeling efficiency of a BP-conjugated fluorescent probe (AF647-ZOL) in *S. aureus* infected mouse tibiae. Consistent with prior reports [29], the results demonstrated with a fluorophore conjugated BP probe, that bisphosphonates efficiently labels all modeling (growth plate) and remodeling (endosteal and periosteal) bone surfaces in the growing animals. In particular, label was observed in higher concentrations at the highest bone turnover sites (Figure 5A). Moreover, Brown–Brenn staining of *Staphylococcus* abscess communities (SAC), immunohistochemistry, and SEM for *S. aureus* confirmed AF647-ZOL incorporation on cortical bone surfaces in immediate proximity to the bacteria (Figure 5B–F).

### 3.2. In Vitro Biofilm Eradication Efficacy of Conjugated BP and HBP Conjugated Sitafloxacin and Tedizolid

As an initial screen towards a lead candidate drug, we assessed the ability of BP and HBP conjugated sitafloxacin and tedizolid, versus their parental antibiotic controls, using an established in vitro biofilm eradication assay, as described in Figure 4. SEM analyses of MRSA static biofilms on the HA discs confirmed the cytolytic activity of the conjugated antibiotics, as evidence by the striking appearance of the shrunken, porous, decomposing bacteria in the drug treated samples versus the BP controls (Figure 6). Quantitative analyses confirmed the great potency of both sitafloxacin conjugates (*O*-phenylcarbamate linkage) on MSSA static biofilm, and no CFU could be recovered even at the lowest dose (1 mg/L) of both HBCS and BCS, while parenteral sitafloxacin was only effective up to 5 mg/L (Figure 7A). In contrast, the BP-tedizolid conjugates (N-phenylcarbamate linkage) (HBCT and BCT) displayed decreased efficacy versus parental tedizolid (Figure 7B). Reasoning that this might be due to the more stable *N*-phenyl carbamate linker, such that the antibiotic was not being released after HA binding [37], we evaluated tedizolid-conjugated to BP with a more labile ester linker (BET), and found this compound to be highly effective, as no MSSA CFU were obtained at any concentration tested (Figure 7B). While HBCS, BCS and BET were also effective against MRSA static biofilm, these BP conjugated antibiotics were only broadly effective at the 800 mg/L dose (Figure 8A,B). BET did have significant activity at the 100 mg/L dose, likely due to its even more labile linker providing higher local concentrations of tedizolid, versus those generated with carbamate linkages. At 100 mg/L, the hydroxy BP HBCS showed better activity against MRSA static biofilm compared to non-hydroxy BP BCS (Figure 8A), which may be attributable to its overall higher binding affinity to bone mineral of HBCS with the additional α-hydroxyl group between two phosphonates [38].

## 
4. Discussion


As a result of its great density from calcification, bone has unique pharmacodynamic challenges for drug targeting, including antibiotic treatments. While the major clinical challenges in curing chronic osteomyelitis are well known, the futility of so called “local” antibiotic therapy achieved from placing a high concentration of drug proximal to the infection in ALBC has become a great topic of debate, as there are no clinical studies that have demonstrated efficacy over systemic antibiotic therapy alone, and the costs of ALBC have been called into question [23]. Moreover, the seminal in vivo rabbit study of Giers et al. [39], which demonstrated that MRI contrast agents similar in size and solubility to common antimicrobials mixed with bone cement are transported away from the bone lesion, suggests that interstitial fluid convection thwarts such “local” antibiotic therapies, as illustrated in Figure 1. Given these issues and the lack of success in treating *S. aureus* bone infection, new approaches are evidently needed to target antibiotics more effectively to bone infection sites.

Bisphosphonates (BP) are a drug class that were specifically developed to target bone with great specificity and minimal side effects on soft tissues [25]. Salient examples are the potent nitrogen-containing bisphosphonates used for the treatment of bone diseases, such as osteoporosis and osteolysis related to metastatic bone [40]. The bone-binding affinity of a BP is predominantly determined by its two geminal phosphonate groups (P-C-P), which form strong bi- and tri-dentate interactions with calcium ions [41,42]. Several different types of bone-targeting drug conjugate candidates have been developed by linking them to BP (e.g., bortezomib [43,44]), and the subject has been extensively reviewed [45,46]. We have sought to create a bone-targeted antibiotic conjugate that could achieve sustained concentrations of drug well above the MBEC at the site of bone infection. Of note, this approach is expected to overcome the challenge of interstitial fluid convection following systemic administration (Figure 1). In support of this concept, we show for the first time that systemic administration of a BP-conjugated fluorophore (AF647-ZOL) probe specifically labels the cortical surface of bone in immediate proximity to *S. aureus* bacteria during chronic osteomyelitis where osteolysis is prolific (Figure 5). As this is a prelude to formal in vivo proof of concept studies with both male and female mice to demonstrate the safety, pharmacokinetics, and efficacy of BP and HBP-conjugated sitafloxacin and tedizolid for the treatment of *S. aureus* osteomyelitis, future studies aimed at assessing prophylaxis, and multiple dosing regimens are planned to assess the potential of this approach to reduce the risks of infection, and if the bone targeted drug accumulates adequately over time. In addition to BP, other bone mineral seeking agents have also been tested preclinically with moderate success. [47,48]. Recently, Rotman et al. showed that microspheres coated with biodegradable polyesters such as poly (aspartic acid) and poly (ε-caprolactone) displayed enhanced bone targeting properties and sustained antibiotics release [49].

Although our initial success with BP-conjugated ciprofloxacin is promising [28], we acknowledge that fluoroquinolones suffer from reduced activity against established biofilms, which is particularly true for virulent strains of *S. aureus* [30,31]. Therefore, this study examined antibiotic conjugates designed to exhibit more effective biofilm eradication. Although we could have used antibiotics usually commonly used to treat bone infection (i.e., gentamicin), we chose to investigate BP-sitafloxacin conjugates on the basis of on our prior FDA-approved drug library screen for bactericidal activity against *S. aureus* SCV. While four library members (daunorubicin, ketoconazole, rifapentine, and sitafloxacin) exhibited potent SCV bactericidal activity against a stable *S. aureus* SCV, only sitafloxacin was potent against MSSA and MRSA established biofilms [50]. Moreover, we showed in murine infected bone defect models that incorporating rifampin and sitafloxacin into 3D-printed calcium-phosphate resorbable scaffold was more antimicrobial and osteoconductive in a single-stage revision versus conventional gentamicin ALBC in a two-stage revision [51]. We chose tedizolid as an alternative antibiotic based on similar biofilm eradication activity [52]. As expected, the *O*-phenyl carbamate sitafloxacin-conjugated BP and HBP demonstrated substantial biofilm eradication potential against prototypic MSSA and MRSA stains (Figure 7 and Figure 8). Importantly, the lack of efficacy observed with *N*-phenyl carbamate linked tedizolid-conjugated BP and HBP, which was overcome by introducing a more labile ester linker (BET), supports the value of our “target and release” local drug delivery strategy (Figure 7B) and warrants future extension to in vivo validation studies.

While these preliminary studies are promising, they have several limitations that should be noted, including the small number of replicates (*n* = 3), which need to be repeated with larger numbers of samples and time courses. We also need to commence in vivo animal studies to assess the potential clinical relevance of the biofilm eradiation activity we observed with these novel compounds. Several of these studies are planned, and the potential of this technology will become known in the near future.

## Figures and Tables

**Figure 1 antibiotics-10-00732-f001:**
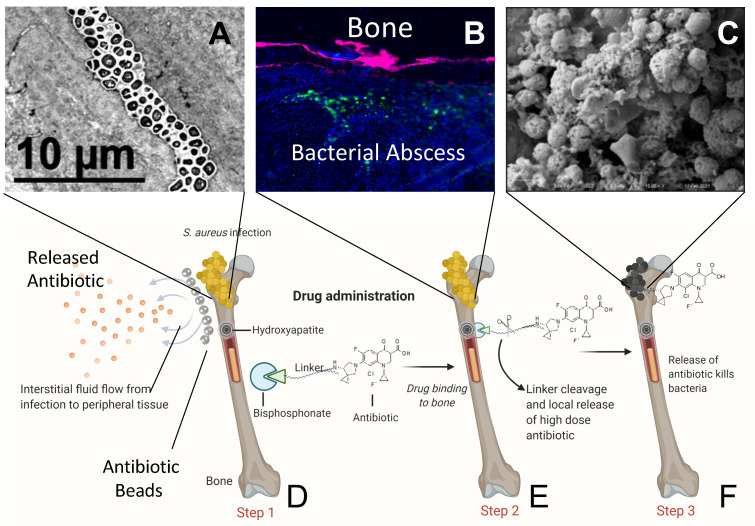
Schematic illustration of bone targeting bisphosphonate-conjugated antibiotic therapy: Theorized bone targeted bisphosphonate-conjugated antibiotics are designed to increase the concentration of systemic (oral or intravenous) or local (antibiotic-loaded bone cement) antibiotics above the minimum inhibitory concentration (MIC) and minimum biofilm eradication concentration (MBEC) levels in immediate proximity to planktonic and biofilm bacteria that have infected bone, (**A**) Transmission electron microscope archival image of *S. aureus* biofilm occupying a canaliculus of the OLCN; (**B**) Bisphosphonate labeled bone (pink) and *S. aureus* positive immunofluorescence labeling (green); (**C**) Scanning electron microscope image of dead *S. aureus* cocci after antibiotic treatment, which cannot be achieved by conventional methods. In this model, the infecting bacteria (e.g., *S. aureus*, gold clustered cocci) have colonized the bone (e.g., proximal femur) and adjacent bone and soft tissues are sheltered from conventional antibiotic therapies by interstitial fluid flow from the wound to peripheral tissues, which drives the drug away from the infection. To overcome these adverse pharmacodynamics (PD), antibiotics conjugated to an inactive bisphosphonate moiety (e.g., (2-(4-hydroxyphenyl)ethane-1,1-diyl)bis(phosphonic acid) (HPBP), blue pie with open wedge) via a linker with adequate serum stability (green wedge with organic chain) that binds with high affinity (10^−6^ M) to hydroxyapatite of metabolically active bone (Step 1) (**D**). Activation of the accumulated bone-bound conjugated antibiotic occurs following linker cleavage via the enzymatic/acidic environment of the infected bone surface (Step 2) (**E**), which releases the drug to kill the bacteria (Step 3, black cocci) (**F**,**C**). For proof of concept, note in (**B**), the fluorescent bisphosphonate labeled bone (red) is found in immediate proximity to a bacterial abscess (green label). (See also Figure 5).

**Figure 2 antibiotics-10-00732-f002:**
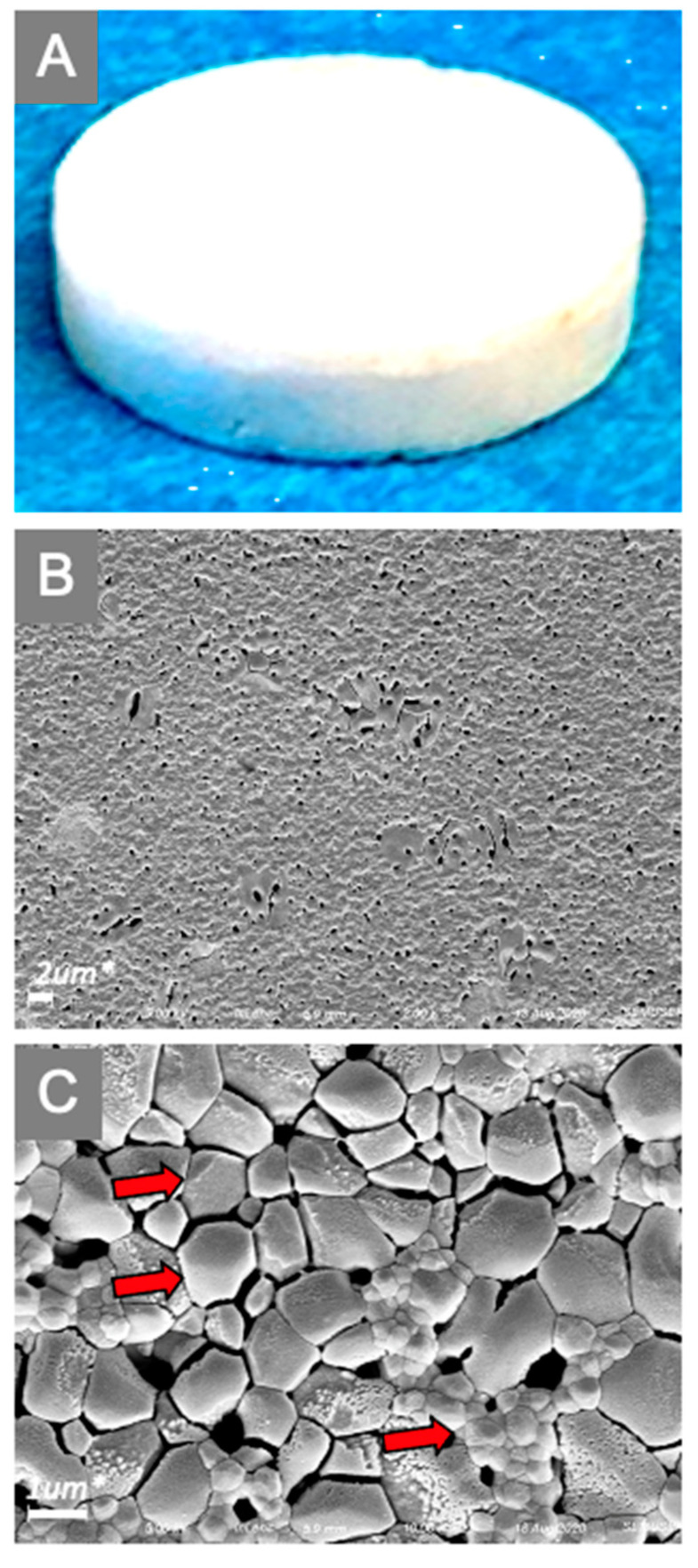
Surface characteristics of experimental HA disc surfaces determined by scanning electron microscope (SEM). Test articles HA discs (**A**) (3D Biotek, LLC. Bridgewater, NJ, USA) received directly from the manufacturer were evaluated by SEM, and representative images of the surface are shown at low magnification (2.00 K×) (**B**), and high magnification (10.00 K×) (**C**). Note the HA disc’s surface has an irregular pebble-like arrangement (indicated with red arrows).

**Figure 3 antibiotics-10-00732-f003:**
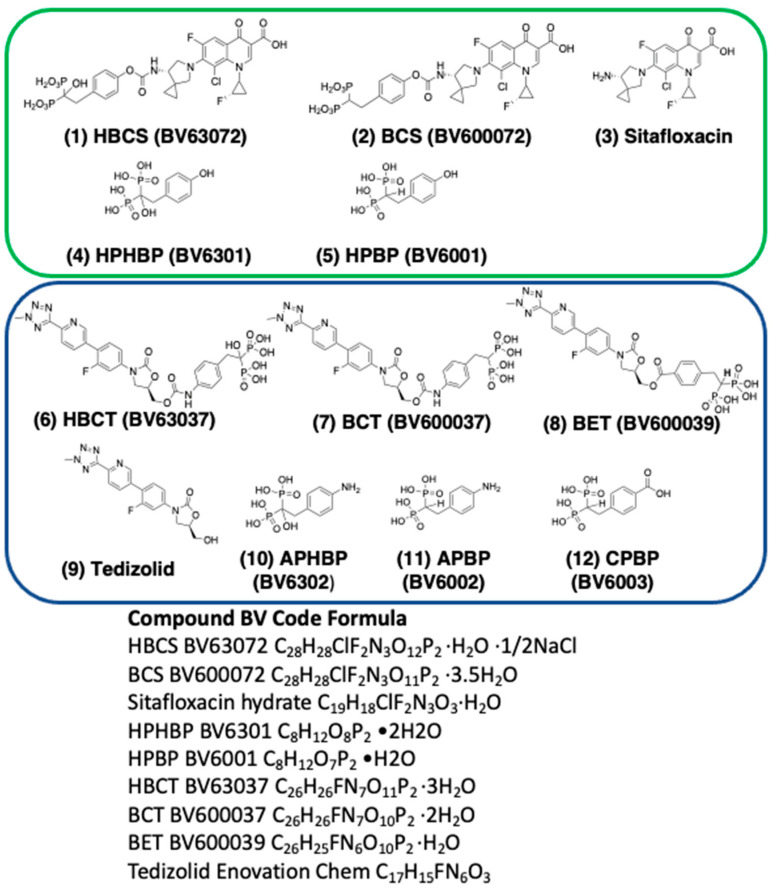
Molecular structure and empirical formula of antibiotics and conjugates: (1) HBCS, (2) BCS, (3) Sitafloxacin hydrate, (4) HPHBP, (5) PHBP, (6) HBCT, (7) BCT, (8) BET, (9) Tedizolid, (10) APHBP, (11) APBP, (12) CPBP were synthesized according to procedures reported by Sun et al. and Sedghizadeh et al. [28,31]. All compounds were characterized by NMR, mass spec, elemental analyses and HPLC demonstrating high purity. On cleavage of the carbamate linkage of (1), (2), (6), (7), the released bisphosphonates (4), (5), (10), (11) and CO_2_ are pharmacologically inactive components. Since cleavage of the linker and release of antibiotic occurs mainly at sites of higher bone resorption caused by osteoclastic or bacterial osteolysis, extremely high local concentrations of antibiotic are generated beneath or in the proximity of infections on bone surfaces. Conjugate (8) utilizes an ester linkage which appears to cleave more readily than carbamate comparators in vitro and will be compared for efficacy in vivo.

**Figure 4 antibiotics-10-00732-f004:**
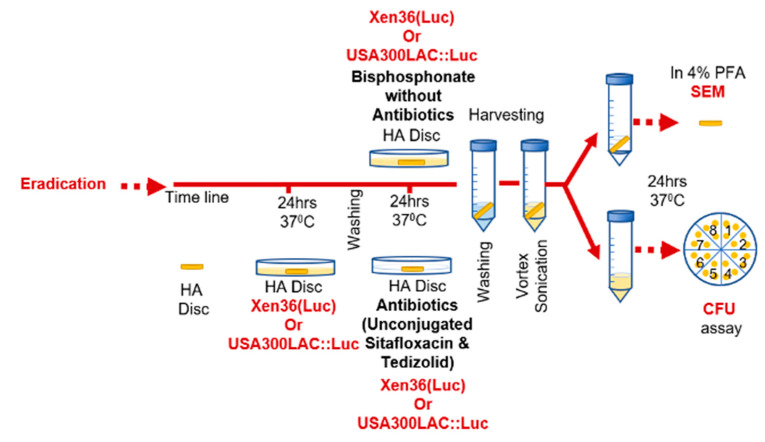
The experimental design of the antibiotic pretreatment to assess inhibition or eradication of biofilm formation on HA discs. 1 mL of tryptic soy broth (TSB) containing 10^7^ Colony-forming units (CFU) of a bioluminescent MSSA strain (Xen36(Luc)) or MRSA strain (USA300LAC::Luc) was added to each well of a 24-well plate, and HA discs were placed in each well and incubated for 24 h at 37 °C After 24 h, HA discs were washed three times in 1 mL of sterile PBS for 5 min in a gentle rocker shaker. Then HA discs were incubated in 800, 400, 200, 100, 50, 25, 10, 5, or 1 mg/L of HBCS, BCS, Sitafloxacin Hydrate, HPHBP, or HPBP, for 24 h at 37 °C. The discs were then rinsed to remove non-bound bacteria, and subjected to vigorous vortex and sonication to collect adherent bacteria from the HA discs. CFUs were quantified via serial dilutions cultured on modified TSB agar plates at 37 °C for 24 h. This protocol was repeated using HBCT, BCT, Tedizolid and BET, HPHBP, or HPBP with concentration of 800 and 100 mg/L.

**Figure 5 antibiotics-10-00732-f005:**
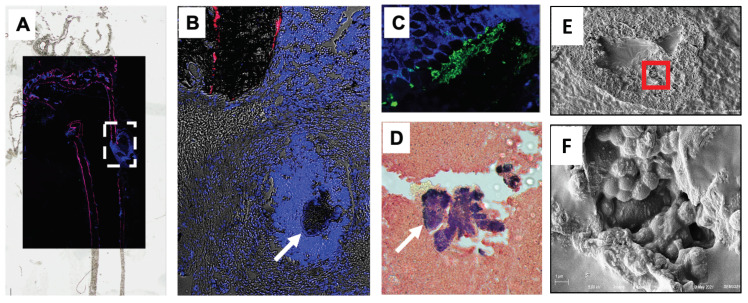
Bone targeted drug labeling of cortical bone proximal to *S. aureus* infection. To illustrate the potential of bone targeted antibiotics, we performed a pilot study in which mice received a transtibial pin contaminated with 10^5^ CFU of MRSA [37], and then received an intraperitoneal injection of AF647-Zoledronate (AF647-ZOL) 7 days later [37]. Mice were euthanized on day 14 post-op, and infected tibiae were processed for undemineralized frozen histology. (**A**) Superimposed darkfield fluorescent image obtained at 4× on top of bright field image of the tibia highlighting the host soft tissue (blue from DAPI stain), and AF647-ZOL labeled cortical bone (red). (**B**) The region of interest (dashed line box in A shown in B at 20×) contains a bacterial abscess in immediate proximity to the labeled bone, which was confirmed to be a *Staphylococcus* abscess community (SAC) (arrow) via immunohistochemistry (green) (**C**) and Brown–Brenn staining (**D**) of serial sections, as we have previously described [37]. (**E**) Scanning electron microscopy image from an adjacent slide to D of the same SAC at 800×. (**F**) Higher magnification of the red box in E at 8000× magnification confirms clusters of *S. aureus* cocci known to reside in SACs that are resistant to standard of care antibiotic therapy.

**Figure 6 antibiotics-10-00732-f006:**
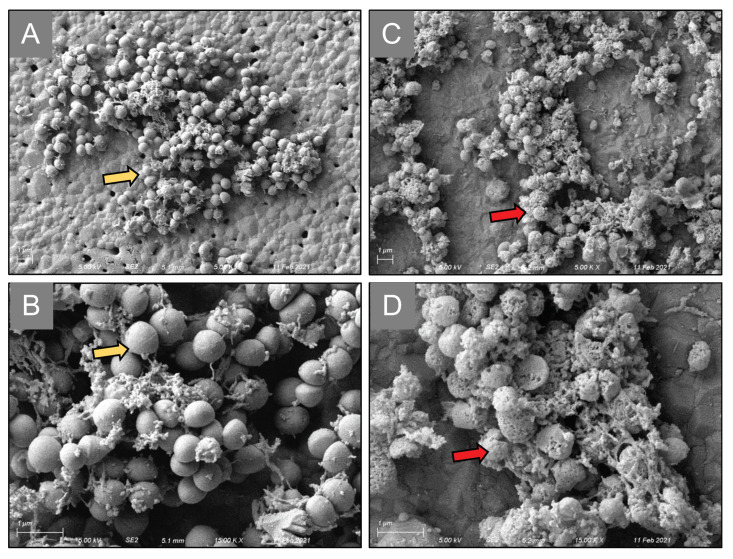
Scanning electron microscopy imaging of *S. aureus* exposed to HBCS antibiotics. Static MRSA biofilms were generated on HA disks and were untreated (**A**,**B**) or treated with HBCS as described in Figure 4. Representative SEM images are shown to illustrate the normal *S. aureus* cells (yellow arrows) from mid-to-late logarithmic growth phase on the HA disc surface at low magnification (5.00 K×) (**A**), and high magnification (15.00 K×) (**B**). In contrast, HBCS treated MRSA static biofilm display a striking appearance of dead bacteria with significant cell wall damage (red arrows) as illustrated at low (5.00 K×) (**C**), and high magnification (15.00 K×) (**D**).

**Figure 7 antibiotics-10-00732-f007:**
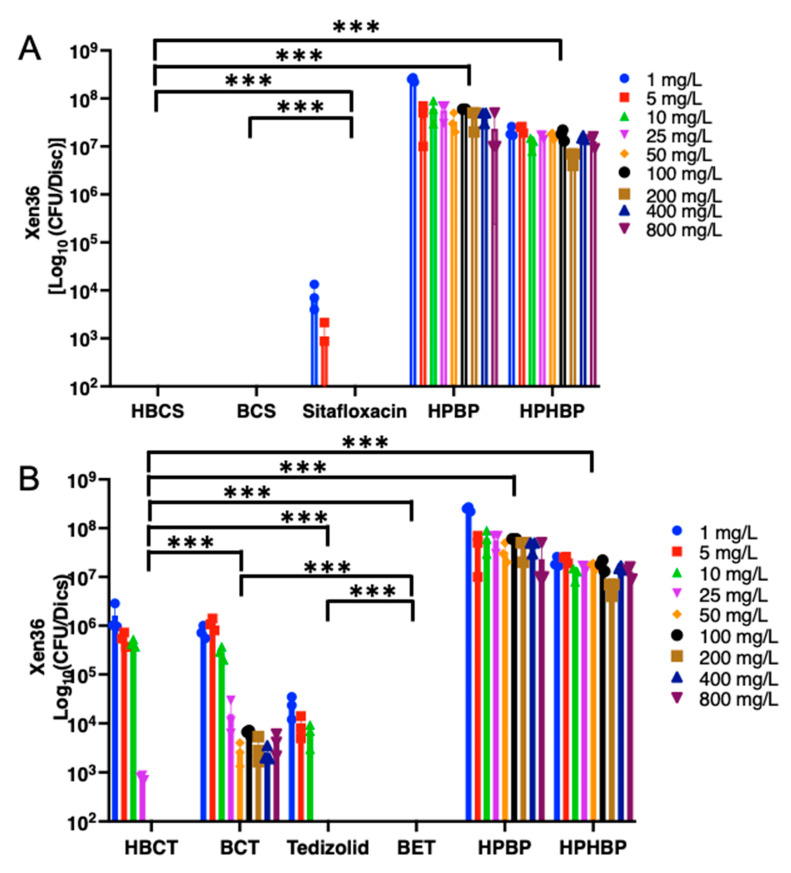
In vitro efficacy of BP conjugated sitafloxacin and tedizolid bone-targeted compounds in MSSA static biofilm assay. Xen36 (MSSA strain) static biofilms were grown on HA discs, and treated with the indicated concentration of (**A**) BP/HBP-sitafloxacin conjugates and Sitafloxacin or (**B**) BP/HBP-tedizolid conjugates and tedizolid, and the control of HPBP or HPHBP and efficacy was quantified via CFU assay as described in Figure 4. Data are presented as the mean +/− SD (N = 3; *** *p* > 0.001 via two-way ANOVA with Tukey’s Multiple).

**Figure 8 antibiotics-10-00732-f008:**
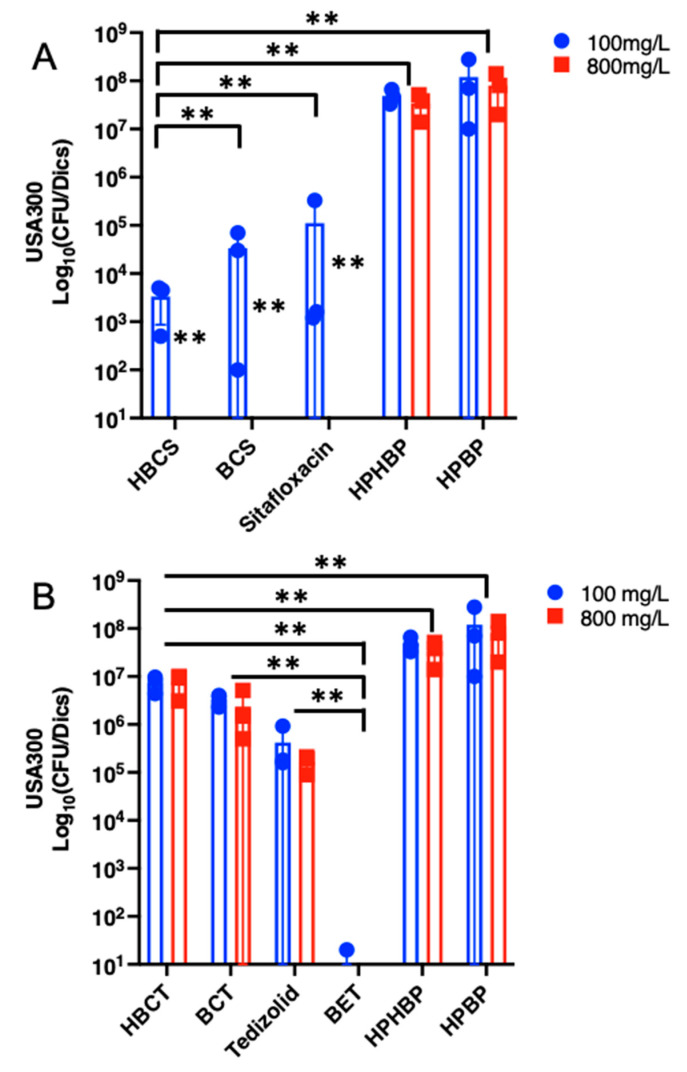
In vitro efficacy of BP conjugated sitafloxacin and tedizolid bone-targeted compounds in MRSA static biofilm assay. USA300 (MRSA strain) static biofilms were grown on HA discs, and treated with the indicated concentration of (**A**) BP/HBP-sitafloxacin conjugates and sitafloxacin or (**B**) BP/HBP-tedizolid conjugates and tedizolid, and the control of HPBP or HPHBP and efficacy was quantified via CFU assay as described in Figure 3. Data are presented as the mean +/− SD (N = 3; ** *p* > 0.05 via Two-Way ANOVA with Tukey’s Multiple).

## Data Availability

The data presented in this study are available on request from the corresponding authors. The data are not publicly available due to patent restrictions.

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
