# Peer review of "Development of Bisphosphonate-Conjugated Antibiotics to Overcome Pharmacodynamic Limitations of Local Therapy: Initial Results with Carbamate Linked Sitafloxacin and Tedizolid"

_antibiotics, 2021, doi:10.3390/antibiotics10060732_

Round 1
Reviewer 1 Report
- What is AF647-Zol treatment? Provide information in material and methods.
- What is the relevance of showing in vivo treatment of treatment AF647-Zol?
- This manuscript is focused on conjugates of sitafloxacin and tedizolid while in vivo experiments using them are missing.
- Also, the authors are saying conjugates sitafloxacin and tedizolid are more effective than the authors should do the comparative efficacy experiments of sitafloxacin and tedizolid with their conjugates.
- Cytotoxicity experiments are missing for sitafloxacin and tedizolid and their conjugates, which authors claiming to be more effective.
- Overall this manuscript has only one relevant experiment (biofilm eradication) for conjugates of sitafloxacin and tedizolid. Unfortunately, the representation of this experiment is not well communicated, and also controls are missing.
- Why only female mice used in the study? What about males? Please justify, that why this study is not biased even if used on female mice.
- Font style is not homogenous throughout the manuscript. Please do so.
- Biofilm formation and treatment section are missing from the material and methods. Please add.
- What about treatment and dosages with conjugated BP and HBP conjugated sitafloxacin and tedizolid? Please add this section to the material and methods.
- Figure7 is very confusing. First of all, this is a simple experiment like MIC determination, so it is strongly recommended to make this figure again in a simple line or bar graph format. What are the positive and negative controls in this figure? The legend should be more detailed with all the information about controls.
- Figure8: Same changes should be made as for Figure4.
Reviewer 2 Report
The manuscript deals with the "Development of Bisphosphonate-Conjugated Antibiotics to Overcome Pharmacodynamic Limitations of Local Therapy".
The manuscript is well written, the experiments well presented, the figures clear and very useful and the results very promising. The conjugation of antibiotics with Bisphosphonate to target bones in which a bacterial infection occurs could be a good strategy to overcome this problem.
I have a few minor comments:
I wonder why the authors did not start with antibiotics usually used against bone infection, like gentamicin.
L156 maybe there is a mistype "indlimbs" should be "hindlimbs"
Author Response
The manuscript is well written, the experiments well presented, the figures clear and very useful and the results very promising. The conjugation of antibiotics with Bisphosphonate to target bones in which a bacterial infection occurs could be a good strategy to overcome this problem. I have a few minor comments:
1. I wonder why the authors did not start with antibiotics usually used against bone infection, like gentamicin.
We thank the Reviewer for this question. To address this we have revised the Discussion to state, “Although we could have used antibiotics usually used against bone infection (i.e. gentamicin), we chose to investigate BP sitafloxacin-conjugates on the basis of on our prior FDA-approved drug library screen for bactericidal activity against S. aureus SCV.”
2. L156 maybe there is a mistype "indlimbs" should be "hindlimbs"
We thank the Reviewer for pointing this out, and have corrected the typo in the revised manuscript.

Round 2
Reviewer 1 Report
The authors successfully response to the reviewer's concerns.